

# The modeled distribution of corals and sponges surrounding the Salas y Gómez and Nazca ridges with implications for high seas conservation

Samuel Georgian[1], Lance Morgan[1] and Daniel Wagner[2]

[1] Marine Conservation Institute, Seattle, Washington, United States
[2] Conservation International, Center for Oceans, Arlington, Virginia, United States of America

## ABSTRACT

The Salas y Gómez and Nazca ridges are two adjacent seamount chains off the west coast of South America that collectively contain more than 110 seamounts. The ridges support an exceptionally rich diversity of benthic and pelagic communities, with the highest level of endemism found in any marine environment. Despite some historical fishing in the region, the seamounts are relatively pristine and represent an excellent conservation opportunity to protect a global biodiversity hotspot before it is degraded. One obstacle to effective spatial management of the ridges is the scarcity of direct observations in deeper waters throughout the region and an accompanying understanding of the distribution of key taxa. Species distribution models are increasingly used tools to quantify the distributions of species in data-poor environments. Here, we focused on modeling the distribution of demosponges, glass sponges, and stony corals, three foundation taxa that support large assemblages of associated fauna through the creation of complex habitat structures. Models were constructed at a 1 km$^2$ resolution using presence and pseudoabsence data, dissolved oxygen, nitrate, phosphate, silicate, aragonite saturation state, and several measures of seafloor topography. Highly suitable habitat for each taxa was predicted to occur throughout the Salas y Gómez and Nazca ridges, with the most suitable habitat occurring in small patches on large terrain features such as seamounts, guyots, ridges, and escarpments. Determining the spatial distribution of these three taxa is a critical first step towards supporting the improved spatial management of the region. While the total area of highly suitable habitat was small, our results showed that nearly all of the seamounts in this region provide suitable habitats for deep-water corals and sponges and should therefore be protected from exploitation using the best available conservation measures.

Corresponding author
Samuel Georgian, samuel.georgian@marine-conservation.org

## INTRODUCTION

The Salas y Gómez and Nazca ridges are two adjacent seamount chains stretching more than 2,900 km off the coasts of Peru and Chile (Fig. 1) (reviewed in *Wagner et al., 2021*).

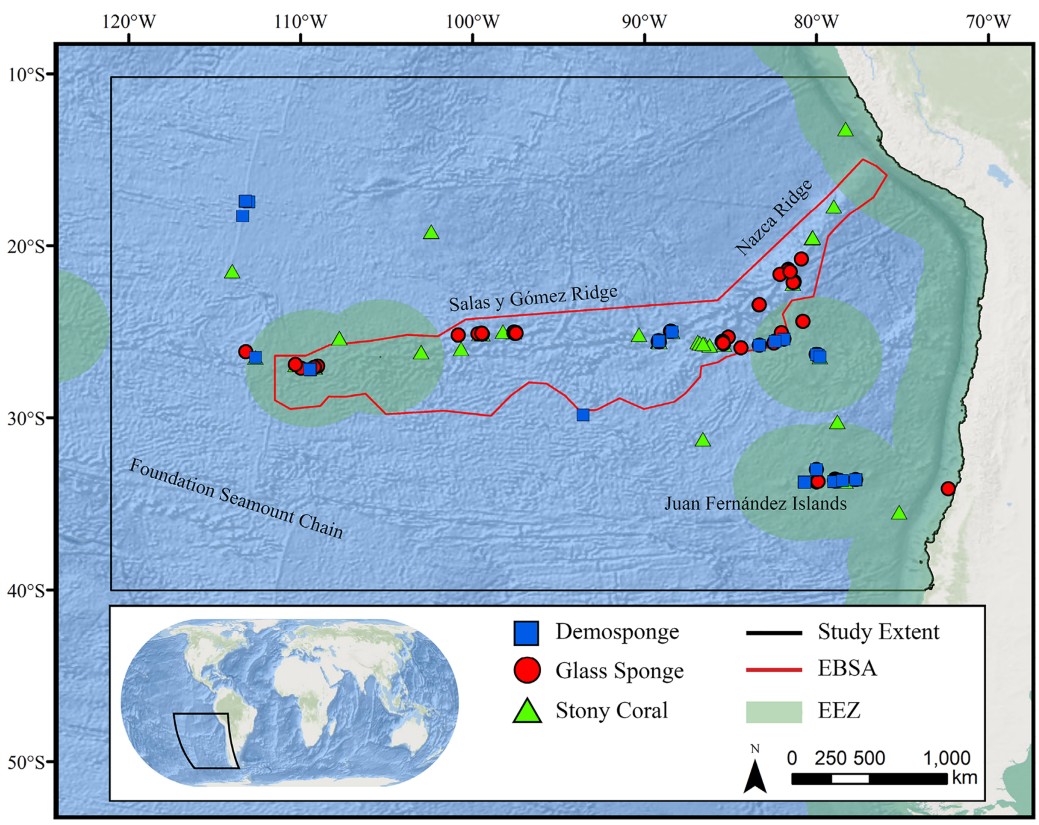

**Figure 1** **Map of the study area.** The map shows the modeling extent, distribution of occurrence records for demosponges, glass sponges, and stony corals, national exclusive economic zones (EEZs), and Ecologically or Biologically Significant Marine Area (EBSA) designation.

Combined, the ridges contain more than 110 seamounts that were created between 2–27 million years ago by a geological hotspot located on the western edge of the Salas y Gómez Ridge (*Parin, Mironov & Nesis, 1997*; *Steinberger, 2002*). The limited exploration that has been accomplished along the ridges has revealed exceptionally high biodiversity as well as unusually high endemism, due in part to its isolation from South America by the Humboldt Current System and the Atacama Trench (*Parin, 1991*; *Comité Oceanográfico Nacional de Chile, 2017*). More than 40% of known fish and invertebrate species are endemic to the region, the highest level of marine endemism in the world (*Parin, Mironov & Nesis, 1997*; *Friedlander et al., 2016*). New species have frequently and recently been discovered on the ridges (*e.g.*, *Andrade, Hormazábal & Correa-Ramírez, 2014*; *Sellanes et al., 2019*; *Shepherd et al., 2020*; *Diaz-Diaz et al., 2020*), indicating that many new species remain to be discovered. The waters surrounding the Salas y Gómez and Nazca ridges provide important feeding grounds and migratory pathways for an array of important species, including billfish, sharks, sea turtles, seabirds and marine mammals (*Weichler et al., 2004*; *Shillinger et al., 2008*; *Yanez et al., 2009*; *Hucke-Gaete et al., 2014*; *CBD, 2017*; *Serratosa et al., 2020*). On the seamounts and neighboring island habitats, diverse benthic communities form around

shallow-water, mesophotic (*Easton et al., 2019*), and deep-water coral and sponge reefs (*Hubbard & Garcia, 2003*; *Easton et al., 2019*; *Friedlander et al., 2021*).

Deep-water corals and sponges are critical foundation species found in every ocean basin. The complex, three-dimensional habitat structures they produce support thousands of associated species including other invertebrates and commercially important fish (*Rogers, 1999*; *Costello et al., 2005*; *Cordes et al., 2008*; *Kenchington, Power & Koen-Alonso, 2013*). In addition to habitat creation, corals and sponges provide other critical ecosystem services including the alteration of local current regimes (*Dorschel et al., 2007*; *Mienis et al., 2009*), carbon cycling and long-term sequestration (*Oevelen et al., 2009*; *Kahn et al., 2015*), and nutrient cycling (*Wild et al., 2008*; *Tian et al., 2016*). Deep-water corals and sponges are also being increasingly used as avenues for research purposes ranging from biomedical research (*e.g.*, *Hill, 2003*; *Müller et al., 2004*) to reconstructing paleoclimate archives of climate change, pollution, and nutrients (*Smith et al., 2000*; *Williams et al., 2006*; *Cao et al., 2007*). The slow growth rates (*Prouty et al., 2011*), extreme longevity (*Roark et al., 2009*; *Fallon et al., 2010*), and life history strategies (*e.g.*, low recruitment; *Doughty, Quattrini & Cordes, 2014*) make these taxa extremely sensitive to anthropogenic disturbance, and the recovery of damaged communities may take many decades, centuries, or even longer (see *Ramirez-Llodra et al., 2011*; *Baco, Roark & Morgan, 2019*). Considering the extreme logistical difficulties and costs associated with restoration efforts in these remote habitats (*Van Dover et al., 2014*), improved conservation measures are urgently needed to protect these fragile ecosystems before long-term damage occurs.

Like most marine biodiversity hotspots, the Salas y Gómez and Nazca ridges are threatened by a variety of ongoing or imminent anthropogenic disturbances, including commercial fishing, marine debris and plastic pollution, seabed mining, and climate change (reviewed in *Wagner et al., 2021*). Despite these threats and the clear biological value of the ridges, protecting their sensitive benthic communities from anthropogenic disturbance is a complex challenge. Over 73% of the ridges are located in areas beyond national jurisdiction (ABNJ), commonly known as the high seas, where no one country has sole management responsibility and hence international cooperation is necessary. While the portions of the ridge located within the Chilean and Peruvian exclusive economic zones (EEZs) have several established marine protected areas (MPAs) (*MPAtlas, 2021*), the high seas portions of the ridges are more loosely regulated by intergovernmental agencies including the International Seabed Authority (ISA), the International Maritime Organization (IMO), the Inter-American Tropical Tuna Commission (IATTC), and the South Pacific Regional Fisheries Management Organisation (SPRFMO), which regulate seabed mining, shipping, and fishing, respectively. Despite ongoing United Nations negotiations to better protect vulnerable marine ecosystems (VMEs) on the high seas (*UNGA, 2007*; *Rogers & Gianni, 2010*), there is no legal mechanism to establish high seas MPAs that are applicable to all States or sectors. Industrial fishing occurs in an estimated 48% of ABNJ, with fisheries pushing into deeper waters each year as stocks deplete in shallower waters (*Visalli et al., 2020*). Commercial fishing in waters surrounding the Salas y Gómez and Nazca ridges has been relatively limited historically (*Wagner et al., 2021*), providing a unique opportunity to protect this diverse region before it is irrevocably
damaged. However, effecting strong protection in ABNJ is difficult due to the lack of clear legal mechanisms, competing interests, and lack of sufficient data in lesser-explored regions (*Gjerde et al., 2021*).

Species distribution models, also referred to as habitat suitability models, are important tools that help characterize the distribution and niche of taxa in data-poor regions. These models can be particularly useful for deep-water taxa on the high seas, where extremely limited surveys have occurred relative to shallow-water coastal areas (*Fujioka & Halpin, 2014*; *Ortuño Crespo et al., 2019*), and data availability is a considerable obstacle to improved conservation management and scientific advancement (*Vierod, Guinotte & Davies, 2014*; *Wagner et al., 2020*). Species distribution models statistically couple the known distribution of species with relevant environmental parameters to predict niche and distribution in unsurveyed geographic regions or under varying environmental conditions (*Guisan & Zimmermann, 2000*; *Miller, 2010*). Quantifying the biogeographic distribution of ecologically important or threatened species is critical for designing and implementing management plans, shaping future research and exploration efforts, and assessing past, present, and future anthropogenic impacts. Increasingly, species distribution models are being developed specifically to inform marine conservation and management (*e.g.*, *Rowden et al., 2017*; *Georgian, Anderson & Rowden, 2019*) or to predict responses to recent anthropogenic disturbances (*e.g.*, *Georgian et al., 2020*). Models have been successfully developed for a large variety of benthic taxa, including global models for stony corals (*Davies & Guinotte, 2011*), black corals (*Yesson et al., 2017*), octocorals (*Yesson et al., 2012*), and gorgonian corals (*Tong et al., 2013*), as well as large-scale regional sponge models (*e.g.*, *Knudby, Kenchington & Murillo, 2013*; *Chu et al., 2019*). Given their status as foundation species, and the frequent classification of these taxa as indicators of VMEs, which SPRFMO and other fishery management organizations are mandated with identifying and protecting (*e.g.*, *Penney, Parker & Brown, 2009*), it is critical to quantify their distribution.

An improved understanding of the spatial distribution of key taxa throughout the Salas y Gómez and Nazca ridges is necessary for the evidence-based conservation of the region. The suitability modeling in this study will inform ongoing efforts to identify and prioritize key conservation targets along the ridges (see *Wagner et al., 2021*), reinforcing the increasingly clear need to protect sensitive benthic fauna in the region from further exploitation and disturbance from anthropogenic sources. In addition to conservation planning, these models will also support future expedition planning, and will improve our understanding of the niche of cold-water corals and sponges throughout the region.

## MATERIALS & METHODS

### Study area

The study area encompassed a large region (15,991,101 km$^2$) of the southeast Pacific Ocean centered on the Salas y Gómez and Nazca ridges off the coasts of Peru and Chile (Fig. 1). This area contains 755 seamounts and guyots covering a total area of 561,452 km$^2$ (3.5% of the total area; geomorphology data from *Harris et al., 2014*). The region includes

an area that has been recognized as an Ecologically or Biologically Significant Marine Area (EBSA) by the Conference of the Parties to the Convention of Biological Diversity (*CBD, 2014*). The EBSA extends around the ridges (Fig. 1) and includes roughly 285 seamounts and guyots covering a total area of 294,225 $km^2$ (17.2% of the EBSA area). The region is bounded on the eastern side by the Atacama Trench, which along with the Humboldt Current System isolates the ridges from South America (*Von Dassow & Collado-Fabbri, 2014*). The Nazca Ridge is comprised primarily of a large plateau, while the Salas y Gómez Ridge is mostly comprised of a series of smaller seamounts, escarpments, and ridge features (Fig. S21). Seamounts and features farther east along the ridges are progressively older and deeper (*Rodrigo, Díaz & González-Fernández, 2014*). Closer to the South American coast, a series of deep-water canyons extends into the Atacama Trench, while farther offshore the terrain is dominated by a series of large spreading ridges as well as smaller seamounts, ridges, and escarpments. The study area is primarily categorized as abyssal, with the Atacama Trench extending into hadal environments and small areas along the coasts, islands, and shallower seamounts extending upwards onto the slope and shelf (Fig. S22).

## Occurrence records

Geo-referenced coral and sponge records were obtained from the Ocean Biodiversity Information System (*OBIS, 2020*), the NOAA Deep-Sea Coral and Sponge Database (*NOAA, 2020*), and records from recent expeditions to the area (J. Sellanes and E. Easton, 2020, unpublished data). All records were obtained as presence-only records, with duplicate records removed prior to analysis. The bulk of records were focused on the Salas y Gómez and Nazca ridges, with another cluster of records in the neighboring Juan Fernández Islands region. We chose to focus on three higher taxonomic groupings that are often key foundation species on seamounts: stony corals (Order: Scleractinia, $n = 233$), demosponges (Class: Demospongiae, $n = 275$), and glass sponges (Class: Hexactinellida, $n = 134$) (Tables S3–S5).

## Pseudoabsence records

Species distribution models are ideally constructed using either presence-absence or abundance datasets (*Winship et al., 2020*). However, obtaining high-quality, true absence data is often difficult or impossible in remote environments, and particularly for deeper-water species. Even when absences are recorded, they may reflect the lack of systematic observations throughout the entire study area rather than true absence (particularly given the narrow field of view of most submersibles or towed camera arrays and similar issues with other sampling techniques such as tows or dredges). Inferring suitable habitat from absence data may also be misleading due to dispersal limitation, biotic interactions, or historical disturbances (*e.g.*, *Hirzel et al., 2002*). Researchers are increasingly developing methods that account for the lack of true absence data by using sophisticated methods to produce better-than-random pseudoabsence or background data (*e.g.*, *Iturbide et al., 2015*).

One of the primary limitations with species distribution models, and especially with presence-only models, is sampling bias in the occurrence data (*Kramer-Schadt et al., 2013*; *Syfert, Smith & Coomes, 2013*). Although often unaccounted for, sampling bias can introduce significant errors into models, affecting both their performance and ecological interpretability (*e.g.*, *Syfert, Smith & Coomes, 2013*). We chose to reduce the effects of sampling bias by creating pseudoabsence data that has the same bias found in the presence data (*Elith, Kearney & Phillips, 2010*; *Huang, Brooke & Li, 2011*; *Fitzpatrick, Gotelli & Ellison, 2013*). To mirror the sampling bias that likely exists in our presence records, we created a two-dimensional kernel density estimate of sampling effort based on the presence locations for each taxon (Figs. S6–S8). Pseudoabsence records ($n = 10,000$) were sampled using this density estimate as a probability grid, resulting in a set of unique, sample-bias corrected pseudoabsences for each taxon.

## Environmental data

Within the study area, a suite of 44 environmental variables known to influence the distribution of corals and sponges were constructed for use in models (Table 1). Bathymetry for the study area were obtained from the SRTM30+ dataset (*Becker et al., 2009*; *Sandwell et al., 2014*) at a resolution of 0.0083° (approximately 1 km) and used in the creation of several additional layers.

A number of terrain metrics were derived from this bathymetry layer to define the shape of the seafloor. Slope, roughness, aspect, general curvature, cross-sectional curvature, and longitudinal curvature were calculated using the ArcGIS (v10.8, ESRI) toolkit 'DEM Surface Tools' (v2; *Jenness, 2004*; *Jenness, 2013*). Slope was measured in degrees and calculated using the 4-cell method (*Jones, 1998*). Aspect represents the compass direction of the steepest slope and was converted to an index of eastness using a sine transformation and an index of northness using a cosine transformation. Curvature metrics assess the likely flow of water across a feature, with positive values generally indicating convex features that cause water to accelerate and diverge, in contrast to concave features where water would be expected to decelerate and converge. Roughness is a measure of topographical complexity, calculated here as the ratio of surface area to planimetric area, with more positive values indicating more complex terrain. The Topographic Position Index (TPI) was calculated using the toolkit Land Facet Corridor Designer (v1.2; *Jenness, Brost & Beier, 2013*). TPI assesses the relative height of features compared to the surrounding seafloor, with positive areas indicating locally elevated features and negative values indicating depressions. TPI is scale dependent, and was calculated at scales of 1,000, 5,000, 10,000, 20,000, 30,000, 40,000 and 50,000 m. Finally, the Vector Ruggedness Measure (*Hobson, 1972*; *Sappington, Longshore & Thompson, 2007*), which calculates terrain heterogeneity, was calculated with a neighborhood size of 3, 5, 7, 9, 15, 17 and 21 using the Benthic Terrain Modeler (v3.0; *Walbridge et al., 2018*).

To complement the suite of terrain metrics, large-scale geomorphological features expected to provide suitable habitat for corals and sponges were obtained from *Harris et al. (2014)*, including seamounts, guyots, canyons, ridges, spreading ridges, plateaus and escarpments. See Fig. S21 for a map of geomorphological features in the study area.

**Table 1 Environmental variables used in model creation.**

| Variable name | Included in final models | Units | Native resolution | Reference |
|---|---|---|---|---|
| Bathymetry | | meters | 0.0083° | *Becker et al., 2009* |
| | | | | *Sandwell et al., 2014* |
| **Terrain Metrics** | | | | |
| Aspect–Eastness | | | 0.0083° | *Jenness, 2013* |
| Aspect–Northness | | | 0.0083° | *Jenness, 2013* |
| Curvature–General | | | 0.0083° | *Jenness, 2013* |
| Curvature–Cross-Sectional | | | 0.0083° | *Jenness, 2013* |
| Curvature–Longitudinal | | | 0.0083° | *Jenness, 2013* |
| Roughness | | | 0.0083° | *Jenness, 2013* |
| Slope | X | degrees | 0.0083° | *Jenness, 2013* |
| Topographic Position Index (TPI) | X | | 0.0083° | *Jenness, Brost & Beier, 2013* |
| Vector Ruggedness Measure (VRM) | X | | | |
| **Geomorphological Features** | | | | *Harris et al., 2014* |
| Seamount | | | | *Harris et al., 2014* |
| Guyot | | | | |
| Canyon | | | | *Harris et al., 2014* |
| Ridge | | | | *Harris et al., 2014* |
| Spreading Ridge | | | | *Harris et al., 2014* |
| Plateaus | | | | *Harris et al., 2014* |
| Escarpment | | | | *Harris et al., 2014* |
| **Benthic Conditions** | | | | |
| Total alkalinity | | $\mu$mol l$^{-1}$ | 3.6 × 0.8–1.8° | *Steinacher et al. (2009)* |
| Dissolved inorganic carbon | | $\mu$mol l$^{-1}$ | 3.6 × 0.8–1.8° | *Steinacher et al. (2009)* |
| Omega aragonite ($\Omega_A$) | X | | 3.6 × 0.8–1.8° | *Steinacher et al. (2009)* |
| Omega calcite ($\Omega_C$) | | | 3.6 × 0.8–1.8° | *Steinacher et al. (2009)* |
| Dissolved oxygen | X | ml l$^{-1}$ | 1° | *Garcia et al. (2013a)* |
| Salinity | | pss | 0.25° | *Zweng et al., 2013* |
| Temperature | | °C | 0.25° | *Locarnini, 2013* |
| Phosphate | X | $\mu$mol l$^{-1}$ | 1° | *Garcia et al. (2013a)* |
| Silicate | X | $\mu$mol l$^{-1}$ | 1° | *Garcia et al. (2013b)* |
| Nitrate | X | $\mu$mol l$^{-1}$ | 1° | *Garcia et al. (2013b)* |
| Particulate organic carbon (POC) | X | g C m$^{-2}$ year$^{-1}$ | 0.05° | *Lutz et al. (2007)* |
| Regional current velocity | | m s$^{-1}$ | 0.5° | *Carton, Giese & Grodsky (2005)* |
| Vertical current velocity | | m s$^{-1}$ | 0.5° | *Carton, Giese & Grodsky (2005)* |
| **Surface Conditions** | | | | |
| Chlorophyll *a* | | mg m$^{-3}$ | 4 km | Aqua Modis (NOAA) |
| Sea Surface Temperature | | °C | 4 km | Aqua Modis (NOAA) |

**Note:**
Not all variables were retained in the final models produced. 'Reference' refers to either the tool used to create the variable (terrain metrics) or the original data source (other variables).

Data describing benthic conditions at the seafloor were obtained from the World Ocean Atlas (v2; 2013), including temperature, dissolved oxygen, salinity, nitrate, phosphate, and silicate. Carbonate data including dissolved inorganic carbon (DIC), total alkalinity,
and the saturation states of calcite and aragonite, were obtained from *Steinacher et al. (2009)*. Current data describing regional horizontal and vertical current velocities were obtained from the Simple Ocean Data Assimilation model (SODA v3.4.1; *Carton & Giese, 2008*). Particulate organic carbon (POC) flux to the seafloor was obtained from *Lutz et al. (2007)*. Raw benthic data layers were transformed to match the extent and resolution of the other environmental variables using the upscaling approach developed by *Davies & Guinotte (2011)*. This upscaling technique incorporates bathymetry data to approximate conditions at the seafloor and has previously been demonstrated to work effectively on both global and regional scales for a variety of data (*Yesson et al., 2012*; *Georgian, Anderson & Rowden, 2019*). The upscaled WOA data (dissolved oxygen, nitrate, phosphate and silicate) were compared to quality-controlled bottom-water bottle data from the Global Ocean Data Analysis Project (GLODAP v2.2019) to assess how much error may have been present in the raw WOA datasets or introduced *via* our upscaling approach (see Fig. S23).

Surface conditions were assessed as chlorophyll *a* and mean sea surface temperature data obtained from the Aqua MODIS program (*Aqua MODIS, 2018*). Both layers were calculated as the mean value from 2002–2016 at a resolution of 4 km, and resampled to match the extent and resolution of the other environmental layers with no additional interpolation.

## Modeling techniques

Models were constructed using four different techniques that have proven successful in modeling the distribution of cold-water corals and sponges: Boosted Regression Tree (BRT), Generalized Additive Models (GAM), Maximum Entropy (Maxent), and Random Forest (RF). For each modeling technique, the sampling-bias corrected set of pseudoabsences ($n = 10,000$) was used in place of either true absences or random pseudoabsences. Each model outputs a habitat suitability score between 0–1, with higher scores indicating more suitable habitat. While often erroneously referred to as the probability of occurrence, this score does not represent a true probability of occurrence in presence-only models due to the lack of true absences and nonsystematic observation of the study area. The refinement of model parameters, final model, model evaluations, and model outputs (*e.g.*, variable importance and response curves) were completed using 'biomod2' (*Thuiller et al., 2016*), 'gbm' (*Ridgeway, 2004*), 'dismo' (*Hijmans et al., 2017*), 'mgcv' (*Wood & Wood, 2015*), and 'randomForest' (*Liaw & Wiener, 2002*) in R (v3.6.1; *R Core Team, 2019*).

Boosted Regression Tree (BRT) models rely on binary splits in a regression-tree structure to define the response of species occurrence or abundance to environmental variables (*Elith, Leathwick & Hastie, 2008*), and have been successfully used to model the distribution of deep-sea fauna (*e.g.*, *Rowden et al., 2017*; *Georgian, Anderson & Rowden, 2019*). The minimum number of trees was set at 1,000, and a Bernoulli distribution of the presence-pseudoabsence data was assumed. Tree complexity was set to three to allow limited interactions between terms.

Maxent (*Phillips, Anderson & Schapire, 2006*) is a machine learning, presence-only modeling algorithm that has been shown to outperform other presence-only models

(*Elith et al., 2006*; *Tittensor et al., 2009*) and even presence-absence models (*Reiss et al., 2011*). Default model settings were used except the maximum number of iterations was increased to 500 to ensure that models converged. In addition, the regularization parameter (default of $\beta = 1$) was experimentally tested using values of $\beta = 1–10$. Regularization is a smoothing function that controls the complexity of models, with higher values resulting in simpler models with fewer parameters. An ultimate value of $\beta = 5$ was chosen for all taxa based on the performance of preliminary models. Previous Maxent modeling of cold-water corals has shown that increasing $\beta$ improves model performance in areas or conditions outside of the training data, essentially by preventing the model from overfitting to its training data (*Georgian, Shedd & Cordes, 2014*).

A Generalized Additive Model (GAM) is a type of generalized linear model that employs smoothing functions for each explanatory variable (*Hastle & Tibshirani, 1986*). GAM has frequently been used to model the distribution and niche of a variety of marine species including corals and sponges (*e.g.*, *Rooper et al., 2014*). A binomial distribution of the presence-pseudoabsence data was assumed. Various types of smoothers and degrees of freedom allowed were explored in preliminary models, resulting in a thin plate regression spline smoothing function with 12 degrees of freedom used for all variables and modeling runs. Testing higher degrees of freedom (ranging from 4–15) resulted in small improvements in model performance but increased computational time, with no significant model improvements above 12 degrees of freedom (in general agreement with the findings of *Wood, 2017*).

Random Forest (RF) models (*Breiman, 2001*) are a classification or regression, tree-based algorithm that relies on a random selection of explanatory variables and an internal bootstrapping metric to produce and then combine a large number of trees. Default parameters were used except the number of trees was increased to 501 and tree depth was limited to a value of ten to prevent overfitting to the training data. Various tree depths (1–20) were investigated in preliminary models, and a tree depth of ten was ultimately selected as it appeared to improve model performance while preventing models from strongly overfitting to the training data. Tuning tree depth appropriately has been shown to improve model performance without significantly affecting computational time (*Duroux & Scornet, 2018*), with larger than default values often yielding the best results (*Segal, 2003*). It should be noted however that other studies have produced better results by limiting tree depth (*Nadi & Moradi, 2019*), suggesting that the correct tuning value may be dependent on the dataset used as well as how other model parameters are tuned in conjunction. The optimal value of 'mtry' was also experimentally altered in preliminary models, however, the default value (the square root of the number of explanatory variables) consistently performed well and was therefore used across all model runs.

Each modeling approach (BRT, Maxent, GAM and RF) is fundamentally distinct and depends on different structures and assumptions. Therefore, each will produce different habitat suitability maps that may reflect tradeoffs in various aspects of model performance, making it difficult to accurately determine which, if any, model type is superior (*Robert et al., 2016*). To create a more robust final model, we generated an

ensemble model for each taxon based on a performance-weighted average of habitat suitability scores from each model type. The BRT, Maxent, GM, and RF model for each taxon were combined using a weighted average of habitat suitability, with weights based on model performance (AUC scores).

## Model testing

Models were tested using a ten-fold cross-validation procedure that randomly partitioned occurrences into 20% test data and 80% training data. Metrics of model performance were calculated for each run and averaged across all ten runs, however, final models were trained using the entire dataset. Model performance was assessed through a combination of Area Under the Curve (AUC) and the true skill statistic (TSS). AUC is a threshold-independent performance measure that in presence-only models indicates the probability that the model correctly ranks occurrences over background locations. A random model has a theoretical AUC of 0.5, and while the maximum AUC is generally unknowable in presence-only models it is always less than 1 (*Wiley et al., 2003*; *Phillips, Anderson & Schapire, 2006*). The TSS metric is similar to the conventionally reported kappa, but is independent of species prevalence as well as the size of the validation dataset (*Allouche, Tsoar & Kadmon, 2006*). TSS ranges from $-1$ to $+1$ with negative values indicating more random performance and positive values indicating better performance. To fine-tune model parameters (see above), the Akaike Information Criterion (AIC) was also used to assess model performance. AIC helps assess the tradeoff between goodness of fit and simplicity of models, and is therefore commonly used for model selection. A similar ten-fold cross validation approach was used to estimate the spatial uncertainty of the models by randomly withholding 20% of occurrence and pseudoabsence data from model construction with replacement between runs. Uncertainty was then assessed as the standard deviation of habitat suitability scores across all ten model runs. This approach does not account for all possible sources of uncertainty, but provides a useful spatial measure of how sensitive the model is to the sampling of occurrence data and the construction of the pseudoabsence dataset.

## Variable selection

The inclusion of highly correlated variables in species distribution models can reduce model performance and make the results more difficult to interpret ecologically (*Tittensor et al., 2009*; *Huang, Brooke & Li, 2011*). Therefore, we employed a variable selection process to refine our original list of 44 environmental layers to a more parsimonious list of less-correlated variables. Variable selection was based on (1) preliminary model testing including predicted variable importance and impact on overall model performance for BRT, Maxent, GAM, and RF models, (2) correlation and clustering among variables, and (3) known biological importance for each taxon. When variables were highly correlated (Pearson's correlation coefficients >0.7) and clustered together (see Figs. S1 and S2), preference was given to the variable that exhibited the best model performance, was less-correlated with other variables, clustered more independently, or was considered to be more biologically relevant.

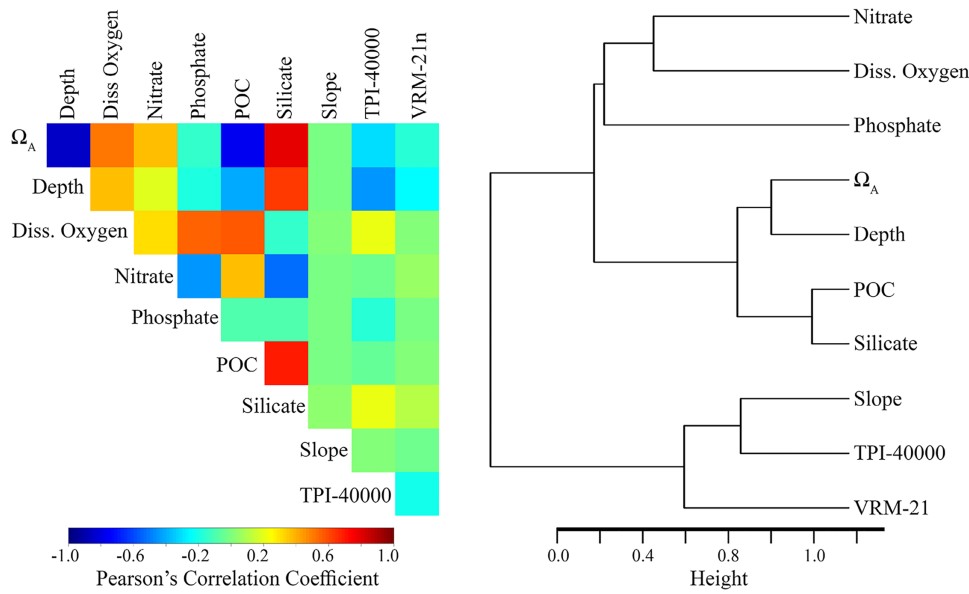

**Figure 2** Relationship among the final set of environmental variables used to train the models. Depth was excluded from the final models but was included here for reference. Silicate was only included for demosponges and glass sponges. $\Omega_A$ was only included for stony corals. Left: Pearson's correlation coefficients among variables. Right: Cluster dendrogram showing the conceptual relationship among variables, with variables containing similar information clustering closer together. See Figs. S1 and S2 for the correlations and clustering of all environmental variables considered for modeling.

Depth was artificially removed despite high model performance due to its high correlation with several, more biologically-relevant variables. The saturation state of aragonite ($\Omega_A$) was included for stony corals due to moderate to high performance in preliminary models and known importance for structuring cold-water coral distributions (*e.g., Georgian, Shedd & Cordes, 2014; Georgian et al., 2016a*). Silicate concentration was included for both demosponges and glass sponges due to the inclusion of silicate in their skeletal structures and high performance in preliminary models. POC was retained despite relatively high correlations with both silicate (−0.684) and $\Omega_A$ (0.769) due to the known biological importance of all three variables and performance in preliminary models. However, it should be noted that ecological interpretations can be difficult when variables are highly correlated.

The final variable set included eight variables for each taxon, including dissolved oxygen, nitrates, phosphates, slope, TPI calculated at a scale of 40,000 m (TPI-40000), VRM calculated with a neighborhood size of 21 (VRM-21), POC, silicate (demosponge and glass sponges only), and $\Omega_A$ (stony corals only). Within the final variable set for each taxon, the highest correlation among variables was −0.684 (POC and silicate), and variables clustered relatively independently compared with the original set (Fig. 2 and Table S1).

## RESULTS

### Model performance

The models performed well across all taxa and modeling algorithms (Table 2). The 10-fold cross-validation procedure produced test AUC scores that were generally above 0.9, with a

**Table 2 Model performance as evaluated by the AUC, and TSS metrics.**

| Taxa | Model | Test data | | Training data | |
|------|-------|-----------|------|---------------|------|
| | | AUC | TSS | AUC | TSS |
| Demosponges | BRT | 0.939 ± 0.02 | 0.783 ± 0.04 | 0.937 | 0.799 |
| | GAM | 0.976 ± 0.01 | 0.912 ± 0.03 | 0.974 | 0.799 |
| | Maxent | 0.975 ± 0.03 | 0.625 ± 0.06 | 0.817 | 0.524 |
| | RF | 0.837 ± 0.01 | 0.932 ± 0.03 | 0.998 | 0.959 |
| | Ensemble | – | – | 0.988 | 0.916 |
| Glass sponges | BRT | 0.868 ± 0.02 | 0.718 ± 0.04 | 0.872 | 0.739 |
| | GAM | 0.904 ± 0.04 | 0.770 ± 0.05 | 0.929 | 0.763 |
| | Maxent | 0.902 ± 0.04 | 0.738 ± 0.06 | 0.904 | 0.679 |
| | RF | 0.948 ± 0.04 | 0.862 ± 0.07 | 0.993 | 0.961 |
| | Ensemble | – | – | 0.968 | 0.770 |
| Stony corals | BRT | 0.915 ± 0.03 | 0.821 ± 0.07 | 0.924 | 0.842 |
| | GAM | 0.956 ± 0.03 | 0.827 ± 0.06 | 0.964 | 0.855 |
| | Maxent | 0.931 ± 0.03 | 0.820 ± 0.06 | 0.939 | 0.822 |
| | RF | 0.965 ± 0.02 | 0.901 ± 0.04 | 0.980 | 0.940 |
| | Ensemble | – | – | 0.973 | 0.845 |

Notes:
Higher values (closer to one) indicate better model performance in each metric. Each metric was calculated using test data during a ten-fold cross-validation procedure withholding 20% of records for testing, and also for the full model using all available training data. Test value are given as the mean ± standard deviation across 10 model runs.

lowest score of 0.837 (RF model for demosponges). Test TSS scores were similarly high, with an average TSS score of 0.809 across all model types and taxa. Test scores did not change considerably among different cross-validation runs, suggesting that ten runs were sufficient to capture the variation caused by withholding different testing data. Glass sponge models performed slightly worse across all modeling types (average test AUC of 0.905 and training AUC of 0.925), followed by demosponges (average test AUC of 0.932 and training AUC of 0.942). Stony corals performed the best by a small margin with an average test AUC of 0.942 and training AUC of 0.952.

Test data revealed that GAM and Maxent models consistently performed slightly better than BRT and RF models. When looking at the evaluation metrics for the final models built using all available training data, RF consistently outperformed the other approaches, consistently having the highest AUC and TSS scores. However, when looking at test scores during cross-validation, RF scores dropped considerably, suggesting that RF models were slightly overfitting to the training data despite optimization of model parameters. In contrast, Maxent models, which generally performed slightly worse than RF models when using training data, had little to no decrease in scores when using testing data, in some cases even performing better against testing data. This suggests that the optimization of the Maxent regularization parameter ($\beta = 5$, default = 1) was successful in reducing model complexity to appropriate levels without a large effect on overall model performance. Testing and training scores were similar for BRT and GAM models, with a very slight decrease in most testing scores. BRT and RF models generally had the lowest

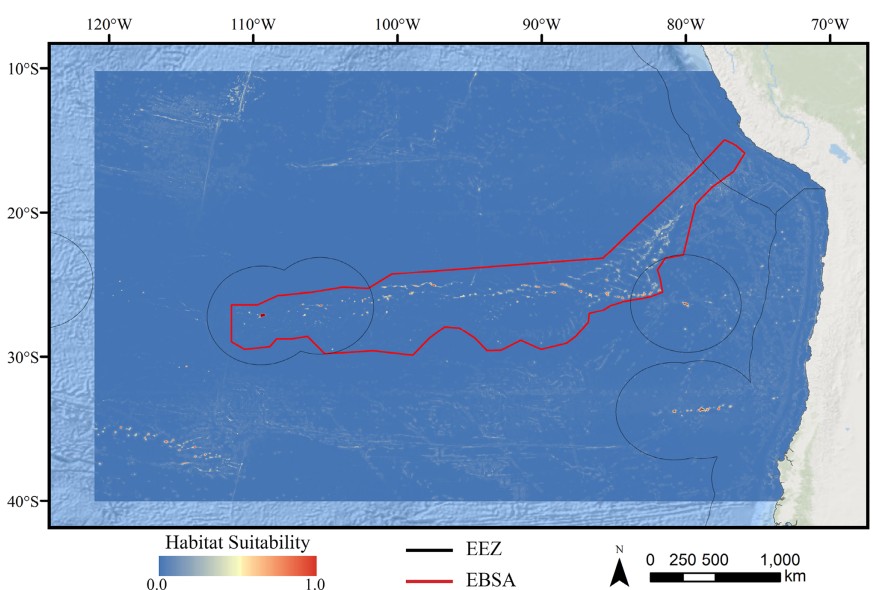

**Figure 3 Predicted habitat suitability for the demosponge ensemble model.** Warmer colors indicate more suitable habitat.

uncertainty (Figs. S24–S35) compared to GAM and Maxent models, although GAM uncertainty was highly spatially restricted to very small areas of highly suitable habitat.

The full ensemble models performed well, with an AUC of 0.988 for demosponges, 0.968 for glass sponges, and 0.973 for stony corals. It is interesting that the demosponge ensemble model performed the best by a small margin, when the individual models performed slightly worse than the models for stony corals. This suggests that the ensemble modeling approach was successful in reducing potential structural inadequacies, lack of ideal model optimization, or bias in each model type that may affect the outputs (*Robert et al., 2016*). In general, the scores for the ensemble models suggest that they outperformed GAM, Maxent, and BRT models, but performed slightly worse than the RF model. However, this is likely because RF models were overfitting slightly, producing artificially elevated training scores. Collectively, the performance metrics suggest that the ensemble models were the best for each taxon.

## Distributions

For all three taxa, areas with high predicted habitat suitability were largely restricted to small pockets clustered around the Salas y Gómez and Nazca ridges, the eastern portion of the Foundation Seamount Chain, and the waters around the Juan Fernández Islands (Figs. 3–5). Glass sponges and stony corals also had strips of low-moderate suitability along the South American coast and along the west flank of the Atacama Trench. Stony corals models also predicted low-moderate suitability on a large spreading ridge along the East Pacific Rise, although there were few highly suitable areas (see Fig. S21). Within the Salas y Gómez and Nazca ridges, suitability predictions were remarkably similar among the three taxa, with highly suitable habitat coinciding with the flanks and summits of most seamount, knoll, and ridge features (see Fig. 6). However, glass sponges appeared to have

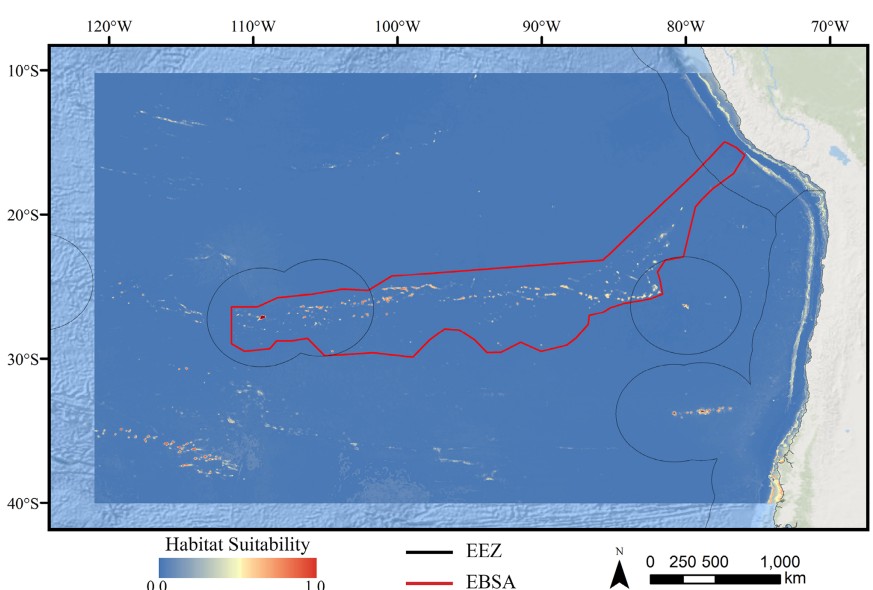

**Figure 4 Predicted habitat suitability for the glass sponge ensemble model.** Warmer colors indicate more suitable habitat.

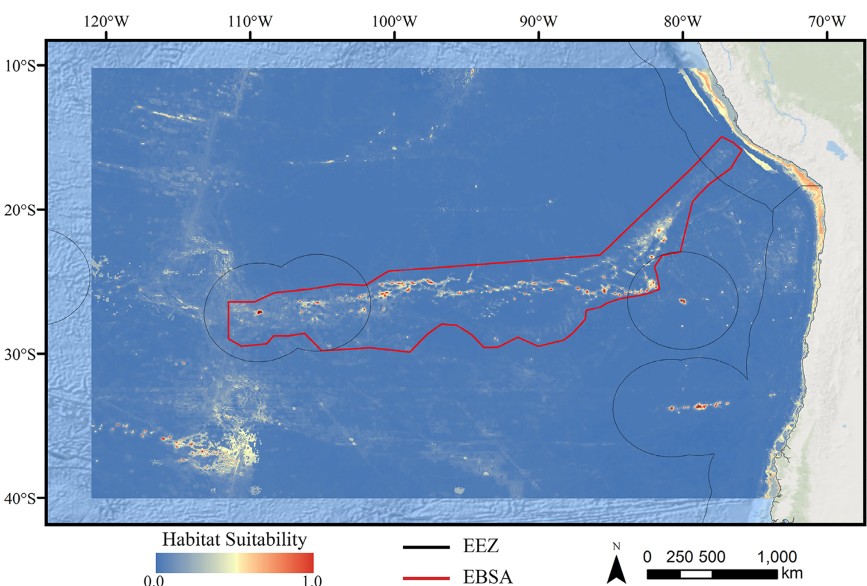

**Figure 5 Predicted habitat suitability for the stony coral ensemble model.** Warmer colors indicate more suitable habitat.

higher suitability predicted on the steeper sides of large seafloor features, while demosponges and stony corals were predicted to occur on the flanks and especially on the summits of the same features.

When the predicted distribution of each taxon was assessed against the large-scale geomorphological classifications (*Harris et al., 2014*, see Fig. S21), a clear preference for escarpments, ridges, seamounts, guyots, and plateaus (primarily along the Nazca Ridge) emerged (Table S2). BRT and RF models typically predicted narrow distributions clustered

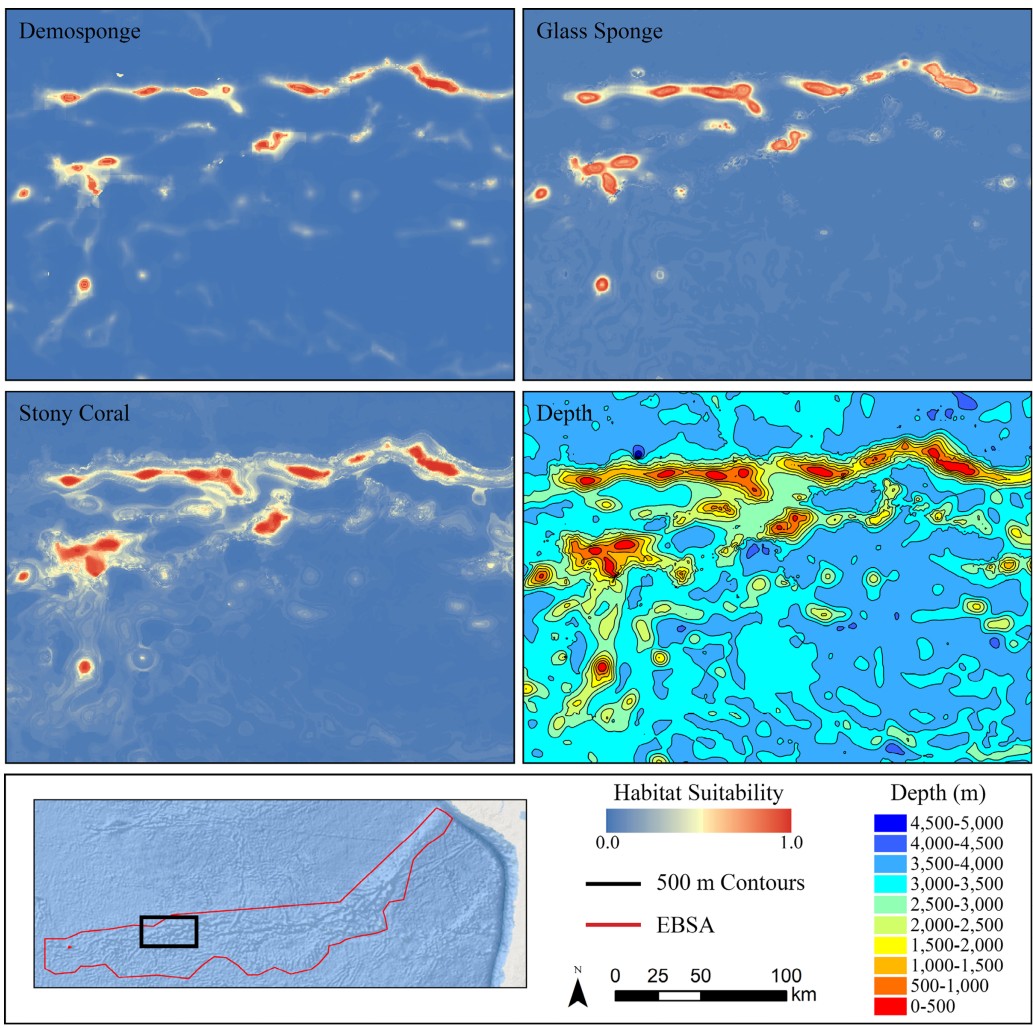

**Figure 6 Ensemble models for demosponges, glass sponges, and stony corals showing a subset of highly suitable seamounts on the western side of the Salas y Gómez Ridge.** Depth with 500 m contours is shown in the last panel for reference.

almost exclusively on seamounts and other large terrain features, while Maxent and GAM also predicted areas of low-moderate suitability in bands along the coast, spreading ridges, and smaller-scale terrain features throughout the region (Figs. S9–S20). While the highly suitability regions were remarkably similar among modeling types, structural differences or assumptions in each model type did affect the overall suitability predictions, lending additional support for the creation and use of ensemble models rather than relying on a single model. It should be noted that model uncertainty was generally highest in areas with higher predicted habitat suitability (Figs. S24–S35), as well as in more coastal areas, suggesting that additional field surveys may improve models in these areas.

## Niche

The niche of each taxa was assessed *via* a combination of variable contribution to the models (Table 3), response curves showing how predicted suitability changes over a range

**Table 3 Percent variable contributions to each model.**

| Taxa | Model | $\Omega_A$ | Dissolved oxygen | Nitrate | Phosphate | POC | Silicate | Slope | TPI–40,000 | VRM–21 |
|---|---|---|---|---|---|---|---|---|---|---|
| Demosponges | BRT | – | 0.0 | 0.4 | 0.3 | **3.7** | **8.5** | 0.1 | **86.0** | 1.0 |
| | GAM | – | **19.0** | 15.9 | 4.2 | **16.7** | **27.4** | 3.8 | 7.4 | 5.6 |
| | Maxent | – | 0.0 | 0.0 | 2.4 | **19.3** | 0.2 | 2.3 | **50.4** | **25.4** |
| | RF | – | 0.6 | 4.5 | 5.8 | 6.5 | **35.5** | 1.9 | **37.4** | 7.7 |
| Glass sponges | BRT | – | 0.1 | 0.3 | 0.1 | **12.5** | **78.3** | **7.3** | 0.6 | 0.9 |
| | GAM | – | 0.9 | **8.5** | **25.7** | 3.3 | **52.5** | 5.0 | 3.5 | 0.6 |
| | Maxent | – | 3.2 | 3.5 | 0.8 | 1.0 | **62.8** | **11.8** | 4.3 | **12.7** |
| | RF | – | 3.5 | 5.7 | 7.9 | **26.8** | **31.1** | **13.6** | 7.5 | 3.9 |
| Stony corals | BRT | 80.3 | 0.1 | 0.1 | 0.0 | 0.4 | – | 0.3 | **18.1** | **0.6** |
| | GAM | 48.9 | 1.0 | 1.6 | **37.4** | **5.2** | – | 0.5 | 2.4 | 2.9 |
| | Maxent | 89.7 | 0.1 | 0.0 | 0.0 | **4.8** | – | 0.3 | 0.0 | **5.1** |
| | RF | 54.1 | 2.2 | **7.0** | 3.2 | 1.6 | – | 1.1 | **28.6** | 2.2 |

Note:
$\Omega_A$ was only included in stony coral models. Silicate was only included in demosponge and glass sponge models. POC=particulate organic carbon. TPI–40,000 = topographic position index calculated at the 40,000 m scale. VRM–21 = vector ruggedness measure calculated with a neighborhood size of 21. The top three variables for each model are highlighted in bold.

of environmental conditions (Fig. 7), and bean plots showing the environmental conditions occurring in the known distribution of each taxa compared to background conditions (Figs. S3–S5).

Terrain metrics consistently contributed a considerable amount of information in each model. Across all model types for demosponges, slope contributed an average of 2% of information, TPI-40,000 contributed an average of 45.3%, and VRM-21 contributed an average of 9.9%. For glass sponges, slope contributed an average of 9.4%, TPI-40,000 contributed an average of 4.0%, and VRM-21 contributed an average of 4.5%. For stony corals, slope contributed an average of 0.6%, TPI-40,000 contributed an average of 12.3%, and VRM-21 contributed an average of 2.7%. Considered jointly, terrain comprised between 15.5–57.3% of information for each taxon, suggesting that the shape of the seafloor is an important component of their niche. Response curves and bean plots suggest that stony corals and demosponges have a clear preference for elevated TPI values (indicating large-scale elevated features). All three taxa show a preference for elevated VRM values (indicating more varied terrain). Interestingly, VRM-21 correlated and clustered with seamounts (Pearson's coefficient of 0.5; Figs. S1 and S2), indicating that at this scale, VRM-21 was a likely an indicator of features similar in topography to seamounts, guyots, and knolls. Bean plots showed that the known distribution of all three taxa coincides with higher slope values of approximately 5–20 degrees compared with background data, however, response curves indicated a more complex relationship. Demosponges exhibited a small preference for elevated slopes, while glass sponges had a large preference for slopes up to approximately 15 degrees, and then a lower preference for steeper slopes. Stony corals had a negative affinity for slopes steeper than 5°.

Silicate was the highest performing variable for glass sponge models (average contribution of 56.2 across model types), and the second highest for demosponges (average
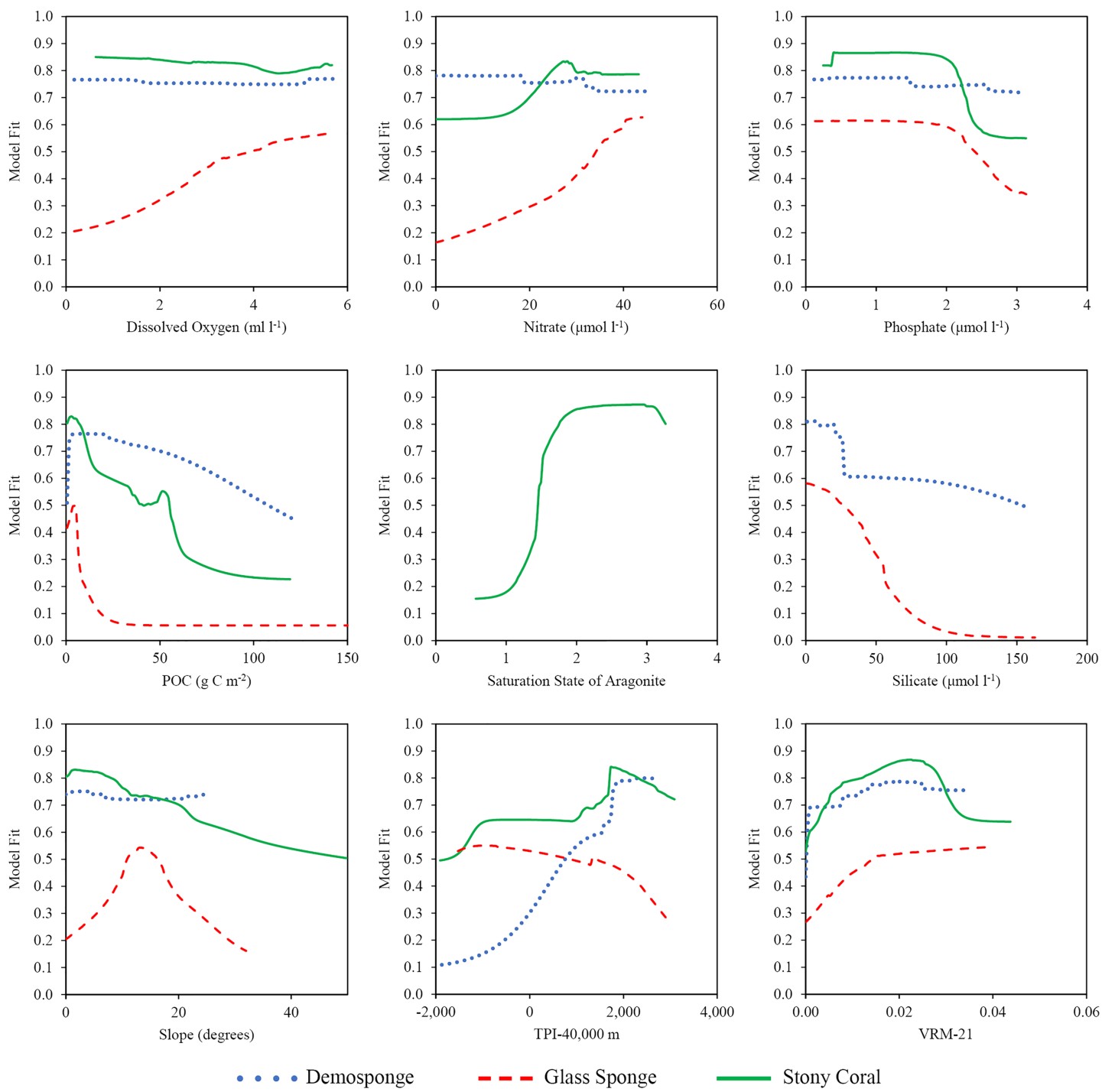

**Figure 7 Response curves showing how the model fit changes over the range of each environmental variable.** Values were calculated for the final variable set using the ensemble model for each taxa. Silicate was only included in demosponge and glass sponge models, and $\Omega_A$ was only included in the stony coral model.

contribution of 17.9% across model types). Response curves indicated that both sponge taxa have a large drop-off in predicted suitability once silicate values exceed approximately 30–40 μmol l$^{-1}$, and bean plots show that both taxa occur at lower than expected silicate

values relative to the background environment. It should be noted that silicate had high correlations with other retained variables, including POC (Pearson's coefficient of $-0.684$) and nitrates (Pearson's coefficient of $0.538$), which complicates the ecological interpretation of these results.

For stony corals, $\Omega_A$ contributed by far the most information in each model type with an average of 68.3% across all models. A clear preference in both response curves and bean plots for elevated $\Omega_A$ values above 1.5 indicated a need for a supersaturated environment. All three taxa were also moderately-highly influenced by POC, with response curves indicating a small spike in suitability between 0–10 g C m$^{-2}$, and then a rapid decrease in suitability at higher concentrations. However, an analysis of the bean plots indicates that POC values $> 5$ g C m$^{-2}$ are rare in the study area, and that all three taxa occur at higher than expected concentrations compared with background values.

Dissolved oxygen, nitrate, and phosphate concentrations were only moderately important in the models for each taxon, generally only entering the top three variables in GAM models, which typically included more moderate contributions from all variables rather than receiving large contributions for a few variables. Response curves indicated that glass sponges preferred elevated dissolved oxygen concentrations, while demosponges and stony corals did not have a clear response. Both sponge taxa appeared to moderately prefer higher nitrate values, while stony corals indicated a small preference for lower values. All three taxa exhibited a moderate preference for lower phosphate concentrations.

# DISCUSSION

## Overview

In order to better inform the spatial management of fisheries and other human activities in the Salas y Gómez and Nazca ridges, we developed ensemble species distribution models for three taxa that are frequently classified as indicator taxa for VMEs (*e.g.*, *Penney, Parker & Brown, 2009*; *Parker, Penney & Clark, 2009*: demosponges, glass sponges, and stony corals). These taxa act as critical foundation species in deep waters due to their habitat creation and other critical ecosystem services (*Roberts et al., 2009*). These areas are VMEs due to their susceptibility to disturbance based on the fragility, rarity, functional significance, and life history traits of their members (*FAO, 2009*). The United Nations require that states and associated intergovernmental agencies identify and protect VMEs, including the closure of fisheries when necessary (*UNGA, 2007*). A better understanding of the spatial distribution and niche of key VME taxa is a critical step towards enacting protection for these fragile and diverse habitats.

The models show that only a small portion of the total study area contained moderately or highly suitable habitat, with the most suitable habitat clustered around topographic highs along the Salas y Gómez and Nazca ridges, the waters around the Juan Fernández Islands, and the Foundation Seamount Chain. The patchy nature of the predicted distribution of all three taxa highlights the difficulties in achieving optimal spatial management with limited observation data, and reinforces the need for species distribution modeling to fill in key knowledge gaps. While the total area of highly suitable seafloor was predicted to be small, these patches extend over large distances, necessitating a regional

conservation approach. It is also important to note that most large-scale features (*e.g.*, seamounts, guyots, ridges, and escarpments) were predicted to be highly suitable for all three taxa, particularly within the Salas y Gómez and Nazca ridges. Surveys in the area have shown that seamounts along the ridges have unique assemblages, including species not found elsewhere along the ridges (*Comité Oceanográfico Nacional de Chile, 2017*), further supporting the argument that protecting all of these features should be a high priority for conservation.

## Influence of environmental conditions

Elevated and more complex seafloor topography has long been known to exert a strong influence on the success of many benthic species including corals and sponges (*e.g.*, *Rowden et al., 2017*; *Chu et al., 2019*). As suspension and filter feeders, corals and sponges are heavily reliant on local and regional currents to increase food supply (*Purser et al., 2010*), transport larva (*Piepenburg & Müller, 2004*), and prevent sedimentation of both tissues and benthic surfaces required for recruitment (*Rogers, 1994*). Elevated and complex terrain affect current regimes in ways that can be favorable for the recruitment and success of cold-water corals and sponges (*Masson et al., 2003*; *Bryan & Metaxas, 2006*). Accordingly, all three taxa in our study appeared to have a strong affinity for seamounts, guyots, ridges, and escarpments, with a clear preference for high TPI and VRM values (indicating locally elevated and complex surfaces).

It was surprising that stony corals appeared to prefer flatter surfaces, while both sponge groups preferred steeper slopes. The response of stony corals to slope may be explained by the low variable contribution of slope to the stony coral model, and the larger contribution of other terrain features. However, a closer examination of the habitat suitability predictions around large seafloor features showed that stony corals appeared to prefer the summits of seamounts and flat-topped guyots to their steeper flanks, suggesting that this relationship may reflect a real preference for being on the tops, rather than the sides, of large terrain features. In contrast, highly suitable glass sponge habitats clustered preferentially on the steeper slopes of large features while also showing a preference for steeper slopes. The flanks and summits of seamounts can contain drastically different environmental conditions due to depth gradients, extreme hydrological forces, exposure to oxygen-minimum zones, and the topography and sediment type of the summit (*Clark et al., 2010*). Further observations on a finer scale than achieved here are necessary to confirm and explain this pattern.

The waters surrounding the Salsas y Gómez and Nazca ridges are generally oligotrophic (*Von Dassow & Collado-Fabbri, 2014*; *González et al., 2019*) and oxygen-poor (*Espinoza-Morriberón et al., 2019*), suggesting that these variables could play large roles in determining species distributions throughout the region. Nitrate and phosphate only contributed a low-moderate amount of information to the models for each taxon, although response curves did indicate that both sponge taxa had an apparent preference for higher nitrate and lower phosphate concentrations. However, research suggests that phosphate and nitrate uptake in sponges is negligible and unlikely to significantly limit distribution (*Yahel et al., 2007*; *Perea-Blazquez, Davy & Bell, 2012*).

Dissolved oxygen concentration is frequently suggested as being critically important for cold-water corals (*Dodds et al., 2007*; *Lunden et al., 2014*) and sponges (*Whitney et al., 2005*). However, it generally did not contribute considerable information to the models in this study, and only glass sponges demonstrated a notable increase in suitability in response to higher dissolved oxygen concentrations. For stony corals and demosponges, this may suggest that dissolved oxygen is not a limiting factor in the region, congruent with other work showing that cold-water coral communities can grow successfully even in very low oxygen conditions (*e.g.*, approximately 2.5 ml l$^{-1}$ at deep-water reefs in the Gulf of Mexico; *Georgian et al., 2016a*). Similarly, research suggests that some sponges can tolerate periods of hypoxia, although they do so at the expense of other metabolic functioning (*Leys & Kahn, 2018*). In contrast, low dissolved oxygen concentrations have been suggested to be the primary limiting growth factor for glass sponge reefs in some regions (*Leys et al., 2004*), suggesting that dissolved oxygen may partially limit their distribution on the Salas y Gómez and Nazca ridges.

POC flux to the seafloor represents a proxy for food supply to benthic communities and is critically important for the success of deep-water corals and sponges (*Wagner et al., 2011*; *Kahn et al., 2015*). POC contributed significantly to the models for all taxa in this study, with suitability predicted to be highest at POC fluxes between approximately 5–50 g C m$^{-2}$. It was surprising that habitat suitability for each taxon actually decreased after POC fluxes of approximately 50 g C m$^{-2}$. However, correlations with other variables including depth, dissolved oxygen, temperature, $\Omega_A$, and silicate may explain this trend. POC values greater than 50 g C m$^{-2}$ were also spatially rare within the study region, with higher values almost exclusively occurring in shallower, coastal waters. In the offshore regions containing deep-water coral and sponge habitats, POC flux was extremely low (generally <10 g C m$^{-2}$), suggesting that these communities may be food limited by default. However, corals and sponges may also uptake dissolved organic carbon (DOC) (*Weisz, Lindquist & Martens, 2008*; *Gori et al., 2014*), or receive food from the lateral transport of POC (*e.g.*, *Rowe et al., 2008*), potentially decoupling the relationship between vertical POC flux to the seafloor and food supply.

Stony corals produce their hard skeletons using the aragonite form of calcium carbonate, with this mineral often serving as the foundation of entire deep-water ecosystems. The saturation state of aragonite ($\Omega_A$) represents the tendency for aragonite to form or dissolve in seawater, with values >1 indicating supersaturated waters where formation is thermodynamically favored. In our study, $\Omega_A$ was the dominant contributing variable in each model type for stony corals, with response curves indicating a clear preference for supersaturated $\Omega_A$ values above approximately 1.5. This conforms with numerous field surveys (*Lunden et al., 2014*; *Georgian et al., 2016a*), experimental results (*Georgian et al., 2016b*; *Kurman et al., 2017*), and modeling studies (*e.g.*, *Guinotte et al., 2006*; *Davies & Guinotte, 2011*) suggesting that aragonite supersaturation is a primary requirement for the growth and success of deep-water stony corals. While survival is still possible in undersaturated waters (*Thresher et al., 2011*), there are large energetic costs associated with calcifying under these conditions (*McCulloch et al., 2012*; *Wall et al., 2015*),

which generally require additional resources *via* increased feeding rates (*Georgian et al., 2016b*).

While silicate was important in both sponge models, it was surprising that demosponge and glass sponge suitability was lower in areas with higher silicate concentrations, as both taxa produce extensive silicate skeletal materials (as much as 80% of the dry weight of glass sponges can be made up of silicate; *Chu et al., 2011*). Previous work has found clear links between sponge distributions and silicate concentrations (*e.g.*, *Whitney et al., 2005*; *Howell et al., 2016*), with silicate uptake becoming more energetically costly when environmental concentrations are low (*Krasko et al., 2000*). However, extensive glass sponge reefs have been documented at similar silicate concentrations (approximately 50 µmol l$^{-1}$; *Chu et al., 2011*) that still coincided with predicted high suitability in our study, with response curves indicating a steep decline in suitability in concentrations starting only around 30 µmol l$^{-1}$. This suggests that it is possible that once a minimum concentration of silicate is reached, there is little additional biological benefit to growing in higher concentrations, allowing the relative importance of other variables to become more important. This finding aligns well with previous research suggesting that a lower silicate level of approximately 30–40 µmol l$^{-1}$ may be the lower limit for optimal sponge growth (*Leys et al., 2004*).

It is also possible that the lower native resolutions of the nutrient, POC, aragonite saturation state, and dissolved oxygen datasets (see Table 1) precluded a more important role in the models, as well as potentially complicating their ecological interpretability. In addition, as comparisons of interpolated layers were not perfectly correlated with water-controlled bottle data from GLODAP (Fig. S23), it is likely that the environmental layers used in our study contain small errors, or that these data are temporally variable. These potential sources of error may complicate the ecological interpretation of these variables, especially when variables are already moderately to highly correlated with other variables (whether included or excluded in final models; see Table S1). However, bean plots and response curves generally demonstrated a strong habitat preference for most key variables, suggesting that these variables are more important on regional scales where the effects of small errors should be negligible. Future work should be completed to improve variable resolution and validate these data in order to more accurately assess how these environmental conditions affect the distribution and niche of these taxa throughout the region. Improved datasets would also considerably improve our ability to predict and map the potential shift in suitable habitat under future scenarios expected with ongoing warming, ocean acidification, shifts in primary productivity, and deoxygenation.

## Threats

Anthropogenic impacts to deep-sea environments are increasing at an unprecedented rate and scale (reviewed in *Ramirez-Llodra et al., 2011*). Despite their remote offshore location, the Salas y Gómez and Nazca ridges are not immune to the risks posed by human activities, including commercial fishing, pollution, climate change, and potential seabed mining (reviewed by *Wagner et al., 2021*). Bottom fisheries are frequently cited as one

of the most damaging activities for deep-water coral and sponge habitats (*Watling & Norse, 1998*; *Pusceddu et al., 2014*), given the indiscriminate and destructive nature of the trawls, lines, and other equipment used. Suitable habitat for corals and sponges often overlaps with bottom fisheries due to the strong association of many demersal fish species with seamounts and similar features, as well as with the habitat structures created by corals and sponges themselves (*Baillon et al., 2012*; *Kutti et al., 2014*). The Salas y Gómez and Nazca ridges have been sporadically but not heavily trawled in the past, with a bottom trawl closure for orange roughy enacted by SPRFMO in 2006 (reviewed in *Tingley & Dunn, 2018*). Long-lining and pelagic fisheries do target the ridges, but for most target species, fishing effort in the region has been historically low (*Wagner et al., 2021*). Therefore, this area presents a unique opportunity to implement strong protections before widespread and irrevocable damage occurs.

Like every major marine habitat studied, the Salas y Gómez and Nazca ridges are affected by marine debris pollution including abandoned fishing gear and plastics, with the bulk of materials stemming from high seas fisheries in the South Pacific (*Luna-Jorquera et al., 2019*) or coastal regions *via* the confluence of the Humboldt Current System and the South Pacific Subtropical Gyre (*Thiel et al., 2018*). Plastic pollution alone is estimated to affect more than 97 species in the region through entanglement and ingestion, including fish, sea turtles, seabirds, and marine mammals. While the harmful effects of marine debris are better documented in pelagic species, microplastics have been found to significantly reduce the growth and feeding of deep-water corals, and derelict fishing gear causes physical damage to deep-water reefs and harms mobile fauna *via* ghost fishing (*Chapron et al., 2018*; *La Beur et al., 2019*).

Seabed mining is a new but imminent threat to many deep-sea environments including the Salas y Gómez and Nazca ridges. Deposits of cobalt, manganese, and polymetallic massive sulfides are known to exist on or near the ridges (*Hein et al., 2013*; *Miller et al., 2018*; *García et al., 2020*), leaving this area susceptible to future mining interests. While seabed mineral extraction has not yet occurred, rising demand for minerals coupled with technological advances in mining equipment are rapidly increasing global interest in the mining industry. If allowed to occur, seabed mining will cause widespread and serious harm to sensitive benthic habitats *via* the physical disruption of the seafloor and sedimentation of neighboring habitats (*Van Dover et al., 2017*).

One of the largest threats to most marine habitats is anthropogenic emissions, which are driving unprecedented rates of warming, deoxygenation, acidification, and decreased productivity in deep-sea environments (*Mora et al., 2013*). For coral and sponge habitats along the Salas y Gómez and Nazca ridges that already experience average or seasonally low dissolved oxygen, low $\Omega_A$, high temperatures, or low POC flux, climate change may rapidly render even highly suitable habitats unviable for the long-term survival of these taxa. If the rate of environmental change is faster than species can adapt or acclimate, the distribution of many fauna may be considerably reduced, potentially resulting in widespread ecosystem collapse (*e.g.*, *Ullah et al., 2018*). Climate change will also exacerbate local stressors including fishing and pollution, reducing both the resiliency of ecosystems as well as their ability to recover from disturbances. However, marine protected areas are

increasingly viewed as a viable tool to mitigate the results of climate change (*Mumby & Harborne, 2010*; *Micheli et al., 2012*; *Roberts et al., 2017*).

## Implications for high seas conservation and management

ABNJ, commonly known as the high seas, cover more than 61% of the global ocean surface and 73% of its volume. These remote ocean areas are not only vast, but also critical for sustaining life on Earth, as they contain nearly 90% of the total ocean biomass, produce nearly half of the oxygen, and capture over 1.5 billion tons of carbon dioxide each year (*Van den Hove & Moreau, 2007*; *Global Ocean Commission, 2014*; *Laffoley et al., 2014*). Yet only 1.3% of ABNJ are currently protected within marine protected areas (*MPAtlas, 2021*), despite widespread and rapidly increasing threats. The lack of high seas protections is in large part due to the makeshift legal framework that is currently in place to protect ABNJ (*Molenaar & Elferink, 2009*; *Gjerde et al., 2016*), as well as the lack of awareness that important ecosystems exist within these remote ocean areas. The results of this study indicate that deep-sea corals and sponges, which build the foundation for some of the most abundant and diverse communities in the deep sea (*Rogers, 1999*; *Costello et al., 2005*; *Kenchington, Power & Koen-Alonso, 2013*), are widespread on seamounts and ridges located in high seas waters of the South Pacific. Given the ecological importance of deep-sea corals and sponges, and their high vulnerability to human impacts, areas that host these communities should be protected from exploitation using the best available conservation measures. Regional fishery management organizations that manage fisheries in this region, namely SPRFMO and IATTC, as well as the ISA which manages seabed mining in international waters globally, already have established mechanisms to protect sensitive marine habitats. Commercial fishing in this region has been very limited in recent years, and deep-sea mineral exploration has not occurred (*Wagner et al., 2021*), providing a time-sensitive opportunity to protect this region without significantly impacting those industries. We thus urge member States of SPRFMO, IATTC, and ISA to close this region to fishing and mining activities before it is too late.

## CONCLUSIONS

The scarcity of data concerning the distribution of key habitat-forming fauna represents an obstacle to conservation efforts. We found that highly suitable habitat for demosponges, glass sponges, and stony corals likely occurs throughout the study area, particularly on large terrain features including seamounts, guyots, ridges, and escarpments of the Salas y Gómez and Nazca ridges. When previously visited during the limited expeditions to the area, these taxa were found to support abundant and diverse ecosystems housing a wide array of associated species (*e.g.*, *Comité Oceanográfico Nacional de Chile, 2017*). It is our hope that these models will inform expedition planning and future research, improve our understanding of the niche and distribution of key taxa, and be considered in a science-based spatial management plan for the region. Anthropogenic disturbance in the deep sea is increasing at an alarming rate, making it imperative to enact strong, permanent protection before these communities are irrevocably damaged.

## ACKNOWLEDGEMENTS

We are grateful for species occurrence records provided by Javier Sellanes (Universidad Católica del Norte) and Erin Easton (The University of Texas Rio Grande Valley). In addition, we are grateful to the NOAA Deep-sea Coral Data Portal and the Ocean Biodiversity Information System for their continued efforts to collect and publish taxonomic records, as well as to all of the researchers who have shared their data through these programs.

### Funding

This work was funded through the Coral Reefs of the High Seas Coalition by Conservation International, the Paul M. Angell Foundation, Alan Eustace, and Tom and Currie Barron. There was no additional external funding received for this study. The funders had no role in study design, data collection and analysis, decision to publish, or preparation of the manuscript.

### Grant Disclosures

The following grant information was disclosed by the authors:
Coral Reefs of the High Seas Coalition.
Conservation International.
Paul M. Angell Foundation, Alan Eustace, and Tom and Currie Barron.

### Competing Interests

The authors declare that they have no competing interests.

### Author Contributions

- Samuel Georgian conceived and designed the experiments, performed the experiments, analyzed the data, prepared figures and/or tables, authored or reviewed drafts of the paper, and approved the final draft.
- Lance Morgan conceived and designed the experiments, authored or reviewed drafts of the paper, and approved the final draft.
- Daniel Wagner conceived and designed the experiments, authored or reviewed drafts of the paper, secured project funding, and approved the final draft.

### Data Availability

Data are available at Dryad: Georgian, Samuel (2021), The modeled distribution of corals and sponges surrounding the Salas y Gómez and Nazca ridges with implications for high seas conservation, Dryad, Dataset, DOI 10.5061/dryad.vdncjsxtc.

### Supplemental Information

Supplemental information for this article can be found online at http://dx.doi.org/10.7717/peerj.11972#supplemental-information.

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
