# Peer review of "The modeled distribution of corals and sponges surrounding the Salas y Gómez and Nazca ridges with implications for high seas conservation"

_PeerJ, doi:10.7717/peerj.11972_

## Round 0.1 · original submission · Minor Revisions

You can see that two reviewers have commented on your manuscript. Overall, they appreciated the approach and they agree that the manuscript is well written. However, they both pointed out some minor/moderate issues that need to be addressed or commented, in particular regarding the environmental data used, the pseudo absence sampling, and the tuning of the algorithms used.

Reviewer 1 ·

Basic reporting

no comment

Experimental design

no comment

Validity of the findings

no comment

Additional comments

This manuscript represents a good modelling exercise of three important benthic taxa. The modelling approach is effective as a tool in marine management and the manuscript is well written and structured. I consider that the study is interesting and the results provided are meaningful. Figures and tables are relevant and well labelled and described. I have only minor suggestions and I recommend publishing the work after a revision following the points suggested below:

Material and Methods
Line 159 – 160. It would be interesting to provide a table with the species or taxon and number of records included in each of the Orders modelled. This could give some guidance about the species or taxon that is driving the results in each group that could be added to the discussion and useful to compare with other studies.

Line 179 – 181. This way of creating pseudoabsence records seems an interesting way to mirror the sampling bias. However, I wonder if this method will provide better results than other recently described (e.g. Iturbide et al. 2015; 2018). Perhaps this needs more discussion.

Iturbide, M., Bedia, J., and Gutiérrez, J.M. 2018. Tackling uncertainties of species distribution model projections with package mopa. The R Journal, 10(1), 122. https ://doi.org/10.32614/ RJ-2018-019

Iturbide, M., Bedia, J., Herrera, S., del Hierro, O., Pinto, M., and Gutiérrez, J. M. 2015. A framework for species distribution modelling with improved pseudo-absence generation. Ecological Modelling, 312, 166–174. https ://doi.org/10.1016/j.ecolm odel.2015.05.018

Line 195. Index of northness using a sine transformation. Should this be cosine transformation?

Line 267. It is common to use 4 degrees of freedom in GAMs to avoid overfitting. However, as the authors are including different taxa in each group, these may not show unimodal response to the environmental variables so more degrees of freedom are ok. However, I wonder why 12 degrees of freedom are chosen and not allow the method to estimate them.

Line 271. I do not understand how 501 trees ensure stabilization in RF when the default is 500. I consider that this needs to be clarified.

Line 272. Could the authors provide a reference or more explanation in how limiting the tree depth to a value of 10 can prevent overfitting to the training data? The results (line 348) show that RF models were slightly overfitting. I wonder if it could be recommended to try different values, as done with ‘mtry’ in preliminary models to assess if this parameter would avoid overfitting.

Line 290. Which threshold was used in the calculation of Kappa? Prevalence? If that is the case, how the selection of pseudo-absences could affect this calculation? Perhaps would be enough to present AUC and TSS that are both independent of species prevalence.

I suggest to present uncertainty maps associated to the predicted habitat suitability maps that can help the interpretability of the results and are very useful for conservation and management.

Results
Line 357. … and 0.952 for stony corals. Should this value be 0.973?\

Discussion
Line 506 – 513. How the lower resolution for some of the environmental variables could affect the results? It may be that it is difficult to find a relationship because the resolution is very low?


References
Some references are missing (e.g. line 60, Cordes et al. 2008; table 1 Carton et al. 2005) or need to be checked (e.g. line 65 Hill 2004; line 72 van Dover 2014)

·

Basic reporting

See below/attachment

Experimental design

See below/attachment

Validity of the findings

See below/attachment

Additional comments

See attachment for some graphics that I've used to illustrate one comment. Below are just copy and pasted.

The authors have produced species distribution models for a little studied region off the coast of South America. Their intent is to use such models to assist in the designation of areas that may contain highly suitable habitats for key benthic taxa in the form of demosponge, glass sponges and stony corals. The authors have followed appropriate methodologies for the modeling work, and the manuscript is clear and well produced. The ensemble approach used is essentially the standard one being used in most modern deep-sea species distribution studies, and the range of model types is appropriate. The quality of writing is high, and I have no major comments pertaining to the writing.

I thank the authors for taking such an approach that is well grounded in the literature. I have only a few minor comments (items 1-3) and one moderate (item 4) methodological comment that I believe the authors should at least respond to.

1. Sampling bias approach
It is great to see consideration of sampling bias in this study, and it is becoming increasingly used, so any case studies that incorporate this, over say, random background selection is valuable. I do have some questions on this, which are likely more discussion points outside of the manuscript. Basically, when I explored the supplementary figures, there were some spurious placements of pseudo-absences in some regions (see example below) and I wasn’t clear exactly how the probability grid was created, and how points were placed based upon it. Can you clarify in the methods the exact approach used, i.e. how were the 10,000 points distributed throughout the grid. As it stands, I wasn’t able to fully visualize the approach.

See attachment for images..

2. Who created the environmental data?
I was not 100% clear if some of the environmental layers (not the topographic layers, and surface derived remote sensing layers) were created specifically in this study or were obtained from Davies and Guinotte (2011), which are freely available on Zenodo. Clearly the use of WOA13 supersedes several environmental layers created by then. It is not a critical point in terms of using the data, but the use of sensu in the attribution, to me, states that you replicated the approach and generated all the layers independently, if you used the data from the Zenodo repository where the Davies and Guinotte layers are stored or obtained it directly from Davies and Guinotte, then you should attribute them more appropriately.

3. Old environmental and bathymetric data?
One of the central requirements for species distribution modeling is to use the best available data. For the presence dataset I have no doubt that it represents the most recent data available. However, the bathymetry and the World Ocean Atlas data used in this manuscript are clearly older. These two products have moved on significantly since SRTM30 was released by Becker et al in 2009 and World Ocean Atlas 2013. There are newer products, what I do not know is how different these products are for the modeled region. It is entirely possible that in GEBCO 2020 or SRTM15, that there are new areas of high resolution multibeam incorporated. Likewise, given World Ocean Atlas is now at version 2018, but I do not know if there are new data from this region. I encourage the authors to explore these data to ensure they are using the most up to date environmental datasets.

4. Validation of environmental layers
This point is valid in either situation outlined in item 2 above. I firmly believe that validation of the environmental data is as important as the validation of the species distribution models themselves. In Davies and Guinotte 2011, extensive calibration of the environmental data extrapolation was undertaken, but this was for the global ocean. When moving down to a regional area, we cannot rely upon a global calibration exercise to show the approach is appropriate. Validation must be undertaken, and it is a fairly simple process. It does not strictly need to be independent, as all we are asking, is how well does the environmental data used fit “reality”. Reality in this case, is out best-known understanding of the main oceanographic conditions. I would really like to see this analysis done, as it is needed to provide full confidence in the models that are created.

---

## Round 0.2 · accepted · Accept

Dear Authors,

I've carefully checked your rebuttal and the revised manuscript, and I've seen that you carefully considered all the comments raised by the two reviewers, implementing most of the suggestions in the manuscript, and -in any case- taking into account the points of the reviewers. For these reasons, I think that the manuscript can be accepted for publication in PeerJ.